# Transcriptomic analysis reveals the impact of concurrent, resistance, and endurance training on skeletal muscle

Longfei Zhao[1☉], Huangyan Li[2☉], Dongli Li[1], Li Luo[3]*, Shiliang Hu[1,3]*

1 School of Sports and Health, Guizhou Medical University, Guiyang, China, 2 Sports Department, Civil Aviation Flight University of China, Chengdu, China, 3 School of Physical Education and Sports Science, Soochow University, Suzhou, China

☉ These authors contributed equally to this work.
* luoli@suda.edu.cn (LL); slhu0000@163.com (SH)

## Abstract

The shared and divergent molecular mechanisms underlying skeletal muscle adaptation to different exercise modalities are not fully understood. This study aimed to compare the physiological and transcriptomic responses to 12 weeks of concurrent (CET), resistance (RES), or endurance (END) training in healthy males. While all groups exhibited similar increases in lean body mass, RES and END elicited distinct functional improvements in maximal strength and aerobic capacity, respectively. Notably, CET preserved strength gains comparable to RES but showed a blunted improvement in anaerobic power. Transcriptomic analyses revealed both common and modality-specific signatures. Although the number of differentially expressed genes varied across groups (CET: 392; RES: 17; END: 49), enrichment analyses consistently identified the engagement of extracellular matrix (ECM) organization pathways. Gene set enrichment analysis further demonstrated a universal activation of ECM remodeling and an inhibition of translation initiation processes post-training. Weighted gene co-expression and protein-protein interaction network analyses pinpointed core genes associated with each modality, including *COL1A1/COL1A2* for CET and END, and *SPARC/ASPN* for RES. Regulatory network predictions implicated the *miR-29* family and *JUN* as potential co-regulators of collagen-related genes. In conclusion, this integrated analysis establishes ECM remodeling as a fundamental transcriptional response supporting exercise-induced hypertrophy common to diverse training modalities, while simultaneously identifying distinct gene regulatory networks that underlie their divergent functional outcomes.

## 1 Introduction

Skeletal muscle, a primary organ for locomotion and metabolism, accounts for approximately 30%−40% of human body weight and exhibits remarkable plasticity,

**Data availability statement:** The dataset (GSE137832) analyzed during this study is available in the Gene Expression Omnibus (GEO) repository, https://www.ncbi.nlm.nih.gov/geo/query/acc.cgi?acc=GSE137832. All other data are within the paper and its supplementary information files.

**Funding:** This study was supported by the University Humanities and Social Sciences Research Project of the Department of Education of Guizhou Province (Grant No. 24RWZX026) and the High-Level Talent Research Start-Up Fund Project of Guizhou Medical University (Grant No. 2024008). Author Longfei Zhao is the recipient of both grants. The funders had no role in study design, data collection and analysis, decision to publish, or preparation of the manuscript.

**Competing interests:** The authors have declared that no competing interests exist.

undergoing extensive molecular and structural adaptations in response to different forms of exercise stimuli [1–5]. Based on distinct physiological demands, exercise modalities are often categorized as "endurance" or "strength" oriented, each inducing disparate adaptive responses in skeletal muscle. Endurance training (END), typically involving prolonged, moderate-intensity muscle contractions, enhances overall aerobic metabolic capacity by promoting mitochondrial biogenesis, optimizing fatty acid utilization, and facilitating a shift towards a slow-twitch fiber phenotype [6–9]. In contrast, resistance training (RES) centers on short-duration, high-intensity loading, primarily activating myofibrillar protein synthesis pathways to drive muscle fiber hypertrophy and strength gains, while exerting relatively limited effects on the muscle oxidative metabolic profile [7,8,10–12]. These modality-specific adaptive mechanisms form the classical theoretical framework of skeletal muscle plasticity.

Concurrent training (CET) is an intervention that systematically integrates both endurance and resistance exercise within a single session or periodic training program, aiming to simultaneously improve various aspects of physical fitness, including aerobic endurance, muscle strength, and cardiorespiratory function [13]. This training paradigm holds broad practical significance. In competitive sports, most disciplines require athletes to possess both superior strength and endurance to support overall performance [14,15]. At the public health level, maintaining and improving muscle strength and aerobic capacity are also crucial for supporting daily physical function and promoting long-term health [16]. Therefore, the scientific implementation of CET is important for optimizing athletic performance and enhancing public health.

However, skeletal muscle adaptations to CET may be constrained by the "interference effect." This concept, first proposed by Hickson in 1980, suggests that concurrently performing different exercise modalities may mutually inhibit their respective adaptive gains [17,18]. For instance, compared to single-mode training, CET may result in comparatively smaller improvements in strength and power, although aerobic capacity enhancements typically remain unaffected [19,20]. While previous studies have preliminarily identified some key molecular pathways involved in RES and END, most research has focused on individual training modalities, lacking a systematic comparison of all three within a unified experimental framework. Furthermore, prior work has predominantly focused on isolated signaling pathways or candidate genes, failing to provide a comprehensive, transcriptome-wide delineation of the complex regulatory networks underlying exercise adaptation. To address these gaps, this study utilizes the public gene expression dataset GSE137832 to systematically compare the effects of CET, RES, and END on the skeletal muscle transcriptome in healthy males under controlled experimental conditions. By integrating differential expression analysis, functional enrichment analysis, co-expression network construction, and regulatory network prediction, we aim to elucidate the common and specific molecular responses of skeletal muscle to different exercise training paradigms, thereby providing molecular-level insights for a deeper understanding of the regulatory mechanisms of exercise adaptation and for optimizing exercise prescription.

## 2 Materials and methods

### 2.1 Data collection and preprocessing

Data were obtained from the Gene Expression Omnibus (GEO) public database, specifically dataset GSE137832. This original study enrolled 18 healthy male subjects, stratified by lean body mass and evenly assigned to one of three training groups: Concurrent Exercise Training (CET), Resistance Exercise Training (RES), or Endurance Exercise Training (END). Baseline demographic and anthropometric characteristics of the participants were retrieved from the primary investigation and are summarized in Table 1. All participants underwent a 12-week supervised intervention, with vastus lateralis muscle biopsy samples collected both before (Pre) and after (Post) the intervention, resulting in a total of 36 transcriptomic sequencing samples.

Data processing was conducted in the R environment (v4.4.3). The raw count matrix was read using the DESeq2 package. To minimize noise from lowly expressed genes, genes with raw counts < 10 in fewer than 6 samples were filtered out. The retained gene expression matrix was then subjected to a variance stabilizing transformation (VST) to mitigate the influence of variations in sequencing depth across samples and to improve the comparability of expression data for subsequent differential expression and clustering analyses.

### 2.2 Exercise intervention protocols

Training interventions were conducted over a 12-week supervised period. The RES and END groups trained on three non-consecutive days per week, while the CET group trained six days per week, alternating between resistance and endurance sessions. The resistance protocol targeted whole-body muscle groups (emphasizing leg press, knee extension, and bench press) at intensities ranging from 60% to 98% of one-repetition maximum (1RM), with 3-minute inter-set rest intervals. The endurance protocol involved cycling on a Lode ergometer, incorporating a mix of continuous moderate-intensity training and high-intensity interval training (HIIT) at intensities ranging from 25% to 110% of peak power output. Progressive overload was applied throughout the 12 weeks by adjusting loads, repetitions, and intervals. Detailed training protocols for each modality are provided in Supplementary File (S1 Table. Training program).

### 2.3 Differential expression genes (DEGs) analysis

Paired Pre/Post samples within the CET, RES, and END groups were analyzed separately for differential expression using the DESeq2 package, employing a design formula (~ Subject + Time) to account for individual variation. The comparison was set as Post versus Pre. Results were adjusted using the Benjamini–Hochberg method, with a significance threshold

**Table 1. Baseline Characteristics of the participants.**

| Characteristic | CET (n=6) | RES (n=6) | END (n=6) | P-value |
|---|---|---|---|---|
| Age (y) | 25±4 | 21±4 | 22±4 | >0.05 |
| Height (cm) | 178±7 | 183±10 | 179±5 | >0.05 |
| Body Mass (kg) | 82.3±10.1 | 75.8±11.4 | 74.3±7.2 | >0.05 |
| BMI (kg/m²) | 25.9±3.2 | 22.6±2.3 | 23.1±1.8 | >0.05 |
| Lean Body Mass (kg) | 60.4±6.2 | 60.0±8.6 | 57.6±5.3 | >0.05 |
| 1RM Leg Press (kg) | 272±73 | 242±79 | 239±43 | >0.05 |
| VO$_2$peak (L/min) | 3.37±0.70 | 3.55±0.74 | 3.51±0.37 | >0.05 |
| Wingate Peak power (W) | 884±156 | 867±200 | 805±169 | >0.05 |

**Note:** Values are presented as mean ±SD. **RES**, Resistance Training; **END**, Endurance Training; **CET**, Concurrent Exercise Training; **BMI**, Body Mass Index; **1RM**, One-Repetition Maximum. Statistical analysis was performed using the Kruskal-Wallis test. No significant differences were observed between groups at baseline ($P > 0.05$).

set at $|log_2FC| \geq 0.5$ and padj < 0.05. Significantly DEGs for each training group were obtained, and the complete results were exported for subsequent analyses.

## 2.4 Functional enrichment analysis

To systematically interpret the functional alterations in the skeletal muscle transcriptome induced by different training modalities, enrichment analysis was performed using R packages including clusterProfiler, biomaRt, and org.Hs.e.g.,db. DEGs were first subjected to Gene Ontology (GO) and Kyoto Encyclopedia of Genes and Genomes (KEGG) pathway enrichment analyses. Furthermore, Gene Set Enrichment Analysis (GSEA) was conducted using gene sets from the MSigDB v2024.1 database, specifically the c2.cp.kegg_medicus (KEGG pathways) and c5.go (GO terms) collections. The significance threshold was uniformly set at padj < 0.05, with the Benjamini–Hochberg method applied for multiple testing correction. The Sankey plot was plotted by https://www.bioinformatics.com.cn (last accessed on 10 Dec 2024), an online platform for data analysis and visualization [21].

## 2.5 Weighted gene co-expression network analysis (WGCNA)

WGCNA was used to identify gene modules significantly associated with the exercise interventions across groups. Network construction was performed using the WGCNA R package on the VST-normalized expression matrix. The soft thresholding power was determined via the pickSoftThreshold function, selecting a power of 9 to achieve an approximate scale-free topology fit ($R^2 \geq 0.9$). A signed Topological Overlap Matrix (TOM) was then calculated, and modules were identified using dynamic tree cutting with parameters minModuleSize = 50 and deepSplit = 3. Similar modules (module eigengene correlation > 0.90) were merged using a mergeCutHeight = 0.10. To pinpoint training-related key modules, samples were annotated with six binary traits (CET/RES/END × Pre/Post), and correlations between module eigengenes (MEs) and these traits were calculated. A p-value < 0.05 was considered significant for correlation. Subsequently, paired t-tests (Pre vs. Post) were performed on the MEs of the significant modules to further validate their dynamic changes in response to the different training modalities.

## 2.6 Core gene identification

Candidate key genes were obtained by taking the intersection between the genes from the significant modules identified by WGCNA and the DEGs for each respective training group. These candidate genes were submitted to the STRING database (https://string-db.org; species: Homo sapiens; confidence score threshold: 0.4) to retrieve protein-protein interaction (PPI) relationships. The resulting PPI networks were visualized in Cytoscape 3.10.3. The CytoNCA plugin was used to calculate three centralities: Betweenness Centrality (BC), Closeness Centrality (CC), and Degree Centrality (DC) [22,23]. Genes were comprehensively ranked based on these three metrics, and the top 10 genes were selected as core candidate genes for further GO and KEGG enrichment analysis to elucidate their potential functional and pathway contexts.

## 2.7 Regulatory network analysis

The NetworkAnalyst platform, utilizing the RegNetwork database, was employed to predict transcription factor (TF) and microRNA regulatory relationships for the core genes. A TF–mRNA–miRNA regulatory network was constructed using the 'Minimum Network' mode. A degree filter threshold of 1 was applied to obtain a high-confidence and structurally concise key regulatory module.

## 2.8 Muscle fiber proportion analysis

To assess the potential impact of training interventions on skeletal muscle fiber composition, deconvolution analysis was performed on the VST-transformed expression matrix using the DeconRNASeq R package [24,25]. Based on a published

signature matrix of slow-twitch (Type I) and fast-twitch (Type II) fiber-specific genes, the proportion of Type I and Type II fibers was calculated for each sample, allowing for the comparison of dynamic changes in muscle fiber composition across the different training modes.

### 2.9 Physiological data analysis

Statistical analyses of physiological outcomes were performed using the R computing environment (v4.4.3), consistent with the transcriptomic data processing. To assess baseline homogeneity across the three groups, the non-parametric Kruskal-Wallis test was employed. To evaluate the specific training adaptations within each group, comparisons between Pre- and Post-intervention time points were conducted using the paired Wilcoxon signed-rank test. Furthermore, to compare the magnitude of training adaptations (calculated as $\Delta = \text{Post} - \text{Pre}$) among the three groups (CET, RES, and END), the Kruskal-Wallis test was utilized. All physiological data are presented as mean $\pm$ standard deviation (SD). Statistical significance was set at $P < 0.05$.

## 3 Result

### 3.1 Participant characteristics and physiological adaptations

Baseline demographic, anthropometric, and physiological characteristics of the 18 participants included in the transcriptomic analysis are summarized in Table 1. Consistent with the stratification strategy of the primary study, Lean Body Mass was highly comparable across the three groups. Similarly, no statistically significant differences were observed in baseline body mass or BMI ($P > 0.05$). regarding functional performance, the groups were well-matched for aerobic capacity ($VO_2$peak), maximal lower-body strength (1RM Leg Press), and anaerobic power (Wingate Peak Power). Statistical analysis confirmed that there were no significant differences among the CET, RES, and END groups for any baseline variable ($P > 0.05$), ensuring a homogeneous baseline for the subsequent evaluation of training-induced transcriptomic and physiological adaptations. The raw data supporting these findings are provided in the Supplementary File (S2 Table. Physiological data).

As shown in **Table 2**, the group results were as follows. Body mass increased significantly after training only in the RES group ($p < 0.05$), but there was no significant difference in the $\Delta$ (Post $-$ Pre) among the three groups ($p > 0.05$). Lean body mass increased significantly within all three groups after training ($p < 0.05$), yet the $\Delta$ did not differ between groups ($p > 0.05$). One-repetition-maximum leg press (1RM Leg Press) showed a significant between-group difference in $\Delta$ ($p < 0.05$); the increases in CET and RES were both significantly greater than in END (CET, RES vs. END; $p < 0.05$). For $VO_2$peak, CET and END increased significantly from baseline ($p < 0.05$); the $\Delta$ differed between groups ($p < 0.05$), with post-hoc tests indicating that the increases in CET and END were both significantly larger than in RES ($p < 0.05$). For Wingate peak power, the groups also differed in $\Delta$ ($p < 0.05$); post-hoc comparisons showed that the $\Delta$ in RES was greater than that in CET ($p < 0.05$), and the END group exhibited a significant within-group increase.

### 3.2 Data preprocessing

The raw count data ($\log_{10}$ scale) exhibited substantial heterogeneity in expression distribution across samples: box heights varied considerably (with some spanning nearly four orders of magnitude), whiskers were asymmetrical (the longest covering up to five orders of magnitude), and outlier points were abundant, indicating notable technical noise and measurement variability (**Fig 1A**). Following VST transformation, the data distribution became markedly more uniform: box heights were largely condensed, whisker lengths were shortened, and the number of outliers was substantially reduced. This demonstrates that variance stabilization effectively compressed the dynamic range and mitigated inter-sample variability, thereby enhancing data consistency and reliability for downstream analyses (**Fig 1B**).

**Table 2. Physiological adaptations to 12 weeks of training.**

| Outcome | CET (n=6) | RES (n=6) | END (n=6) | *P*-value |
|---|---|---|---|---|
| **Body Mass (kg)** | | | | |
| **Pre** | 82.3±10.1 | 75.8±11.4 | 74.3±7.2 | — |
| **Post** | 85.3±8.0 | 79.4±12.3$^\$$ | 76.8±7.8 | — |
| **Δ** | 3.0±4.1 | 3.6±2.1 | 2.5±2.1 | >0.05 |
| **Lean Body Mass (kg)** | | | | |
| **Pre** | 60.4±6.2 | 60.0±8.6 | 57.6±5.3 | — |
| **Post** | 63.0±6.1$^\$$ | 62.3±8.3$^\$$ | 59.9±5.6$^\$$ | — |
| **Δ** | 2.6±1.5 | 2.3±1.2 | 2.2±0.8 | >0.05 |
| **1RM Leg Press (kg)** | | | | |
| **Pre** | 272±73 | 242±79 | 239±43 | — |
| **Post** | 328±73 | 307±59 | 251±44$^\$$ | — |
| **Δ** | 56.2±45.7$^\&$ | 64.6±25.3$^\&$ | 11.7±21.4 | <0.05 |
| **VO$_2$peak (L/min)** | | | | |
| **Pre** | 3.37±0.70 | 3.55±0.74 | 3.51±0.37 | — |
| **Post** | 3.74±0.57$^\$$ | 3.47±0.67 | 3.94±0.49$^\$$ | — |
| **Δ** | 0.4±0.2$^\#$ | −0.1±0.3$^\&$ | 0.4±0.3 | <0.05 |
| **Wingate peak Power (W)** | | | | |
| **Pre** | 884±156 | 867±200 | 805±169 | — |
| **Post** | 913±115 | 869±189 | 990±189$^\$$ | — |
| **Δ** | 29.5±112.7$^\#$ | 123.3±51.1 | 64.8±40.9 | <0.05 |

**Note:** Values are presented as mean ±SD. **CET**, Concurrent Exercise Training; **RES**, Resistance Exercise Training; **END**, Endurance Exercise Training. **Δ** represents the absolute change (Post – Pre). *P*-value indicates the significance of differences in **Δ** among the three groups (Kruskal-Wallis test). $^\$$Indicates a significant difference from Pre-training within the group (*P*<0.05, Paired Wilcoxon signed-rank test). $^\#$Indicates a significant difference in **Δ** compared to the **RES** group (*P*<0.05, Dunn's post-hoc test). $^\&$Indicates a significant difference in **Δ** compared to the **END** group (*P*<0.05, Dunn's post-hoc test).

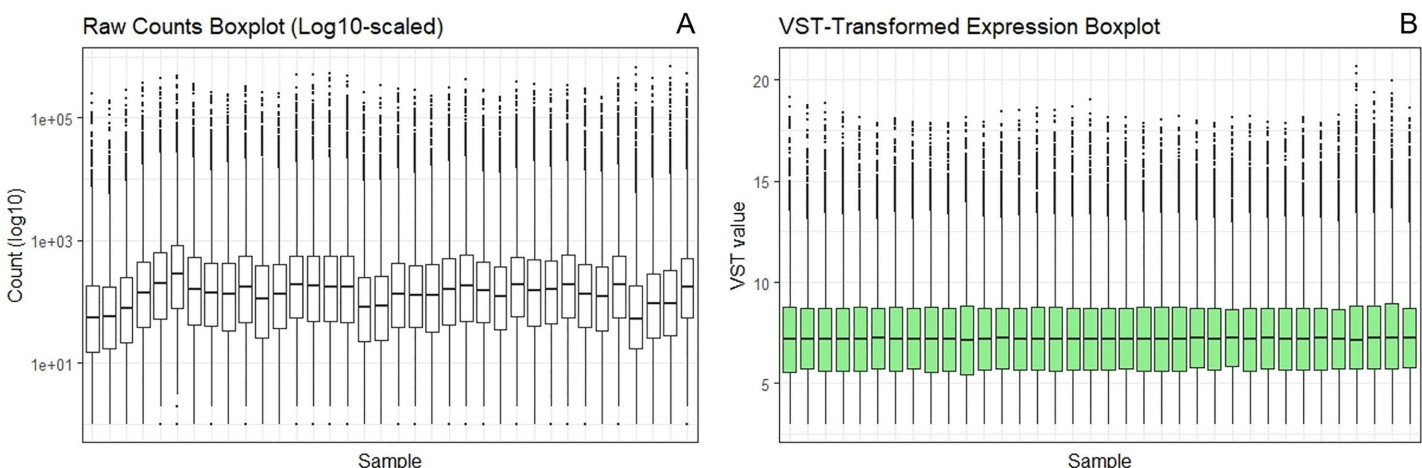

**Fig 1. Boxplots of raw counts and VST-transformed expression data.** Panels A and B display the boxplots of raw counts (log$_{10}$-scaled) and VST-transformed expression data, respectively. The boxplots depict the data distribution across samples, represented by the interquartile range (box), median (central line), whiskers, and outlier points.

## 3.3 DEGs analysis

In the CET group, a total of 392 DEGs were identified (306 upregulated and 86 downregulated). The top 10 genes, ranked by adjusted p-value (padj), are labeled in the volcano plot (Fig 2A), while a heatmap (Fig 2D) displays the relative expression of the top 50 DEGs (ranked by padj). Core upregulated genes included members of the collagen family encoding type I, III, IV, V, and VI collagens (e.g., COL1A1, COL1A2, COL3A1, COL4A1), lysyl oxidase (LOX) involved in collagen cross-linking, and various ECM regulatory proteins (e.g., LUM, OGN, ASPN, THBS4). Concurrently, the upregulation of matrix metalloproteinase MMP14 suggested active ECM remodeling. Among the few downregulated genes, myostatin (MSTN) was the most significantly suppressed.

The RES group yielded 17 DEGs (4 upregulated, 13 downregulated). The top 10 genes (by padj) are annotated in the volcano plot (Fig 2B), and a heatmap of all DEGs is shown in Fig 2E. Upregulated genes included ASPN, SPARC, MYH6, and PTGDS. Downregulated genes encompassed the fast-twitch fiber marker MYH1, the actin regulatory gene LMOD1, and the immune-related gene C3.

For the END group, 49 DEGs were identified (43 upregulated, 6 downregulated). The top 10 genes (by padj) are highlighted in the volcano plot (Fig 2C), and a corresponding gene expression heatmap is presented in Fig 2F. Consistent

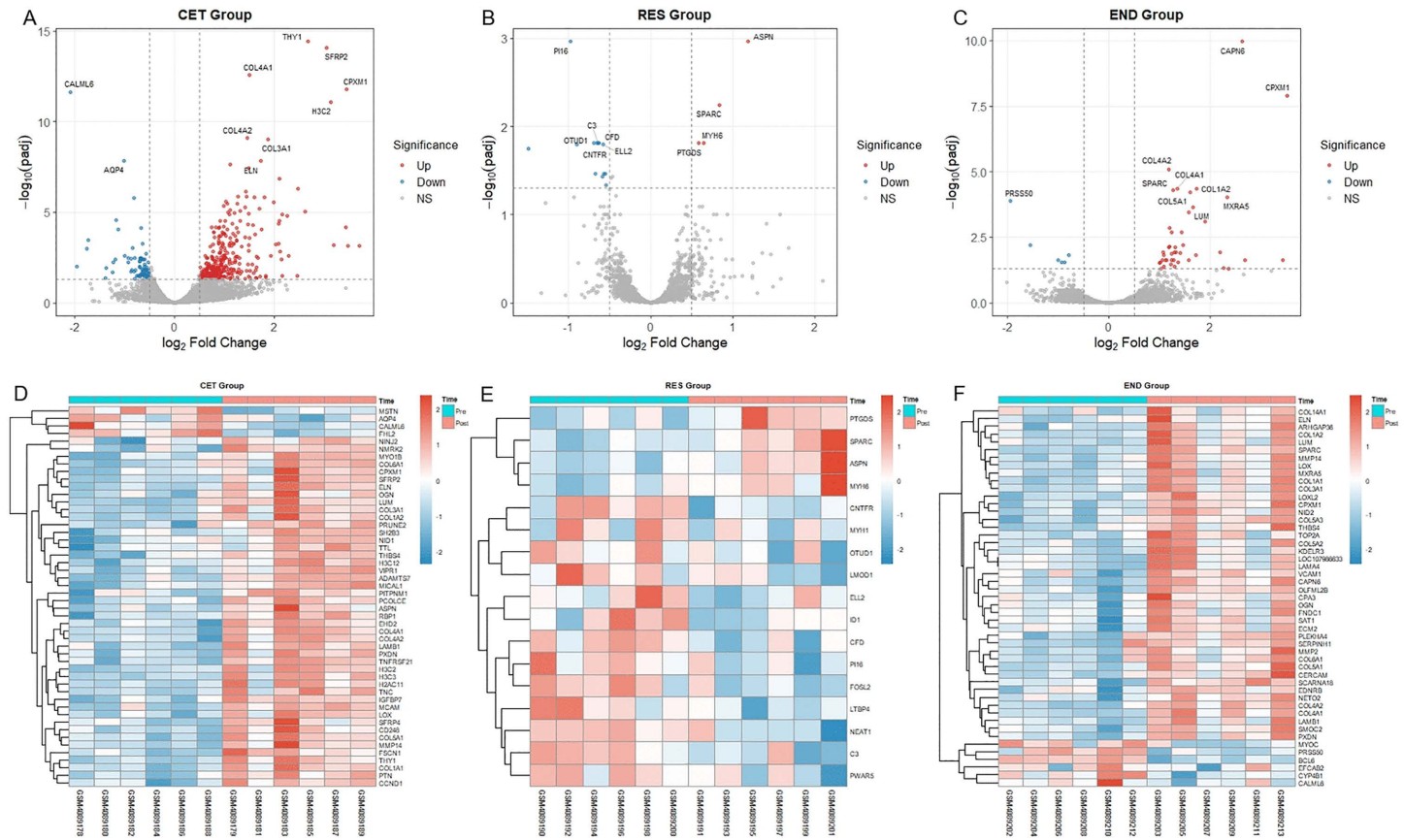

**Fig 2. Volcano plots and heatmaps of DEGs. (A-C)** Upper panels: Volcano plots displaying the distribution of DEGs. The y-axis represents -log₁₀(adjusted p-value), and the x-axis represents log₂(fold change). Red dots denote significantly upregulated genes, blue dots denote downregulated genes, and gray dots represent non-significant genes. The top 10 most significant genes (by adjusted p-value) are labeled in each plot. **(D-F)** Lower panels: Heatmaps illustrating the relative expression levels (Z-score) of DEGs across Pre- and Post-intervention samples. The color gradient from blue to red indicates low to high expression, highlighting distinct expression patterns among the different training groups.

with CET findings, multiple collagen subtype genes were significantly upregulated. Other upregulated categories included ECM regulators and assembly factors (e.g., SPARC, LUM, OGN, THBS4), matrix-modifying enzymes (e.g., LOX, LOXL2, MMP2, MMP14), and cell adhesion and signaling molecules (e.g., VCAM1, LAMA4, LAMB1, NID2). The limited set of downregulated genes, including PRSS50, CYP4B1, CALML6, MYOC, EFCAB2, and BCL6, are implicated in protease activity, metabolic regulation, and transcriptional repression.

To elucidate the potential biological mechanisms underlying the observed differential expression, systematic GO and KEGG functional enrichment analyses were performed on all DEGs. In the CET group (Fig 3A), DEGs were significantly

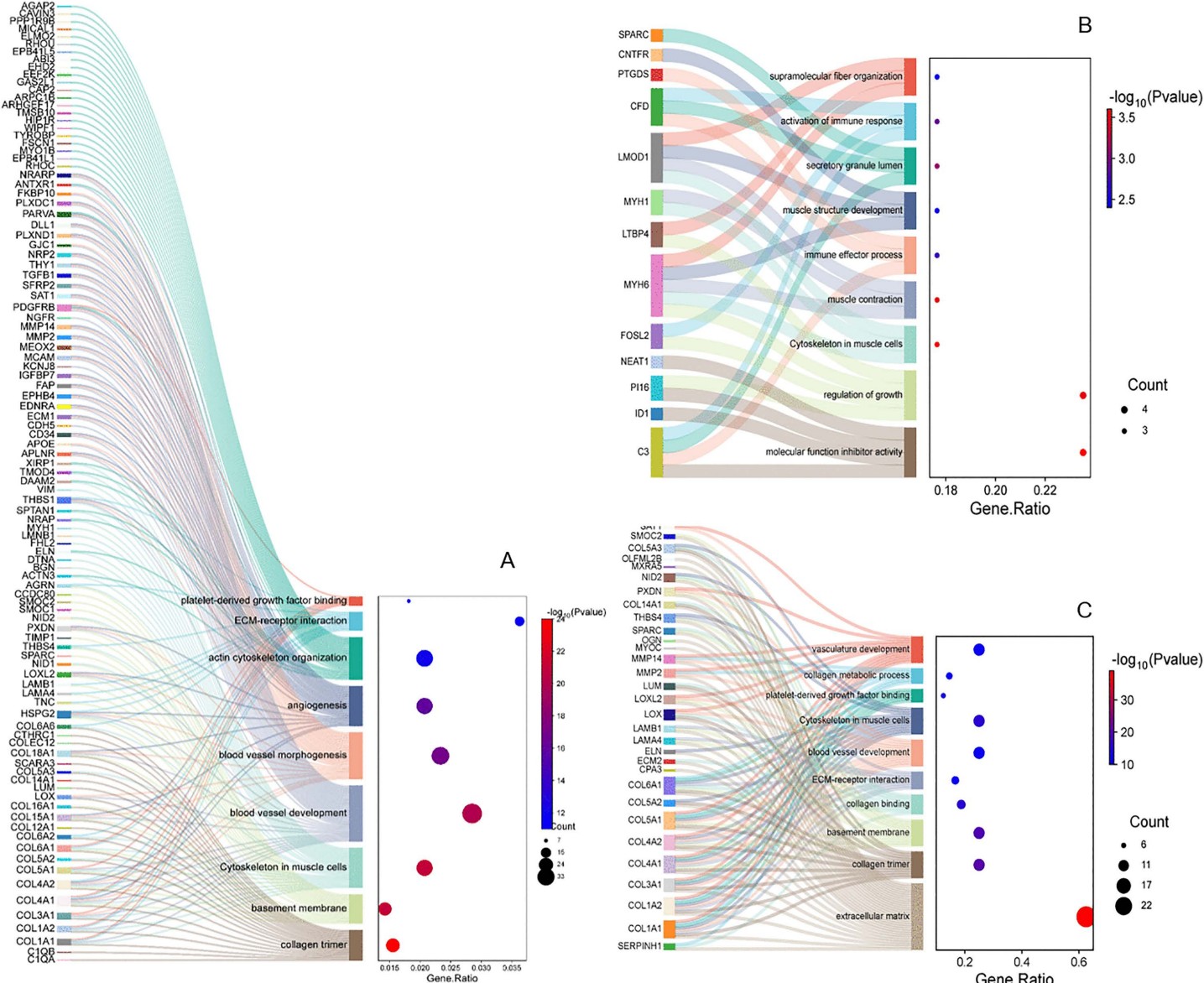

**Fig 3. Functional enrichment analysis of DEGs. (A-C)** Results for the CET, RES, and END groups, respectively. For each panel, the left side presents a Sankey diagram, illustrating the linkages between genes and their significantly enriched pathways. The right side displays a dot plot, where the y-axis represents the enrichment terms, and the x-axis represents the GeneRatio. Dot color corresponds to the statistical significance (-log₁₀(Pvalue)), with a gradient from blue (less significant) to red (more significant). Dot size is proportional to the number of genes associated with each term.

enriched in pathways related to extracellular matrix (ECM) structure, including "collagen trimer", "basement membrane", and "ECM-receptor interaction". Enrichment was also observed in terms associated with "myofibril organization", "vascular development", and "platelet-derived growth factor binding". For the RES group (Fig 3B), enriched functions were predominantly structural, such as "supramolecular fiber organization", "contractile fiber/actin cytoskeleton in myocytes", and "muscle structure development/contraction". Immune-related functions like "immune response" and "immune effector process" were also significant, alongside other regulatory categories including "secretory granule lumen" and "growth regulator activity". The END group (Fig 3C) also showed primary enrichment in ECM-related pathways, such as "collagen trimer", "basement membrane", "collagen binding", and "ECM-receptor interaction", complemented by terms like "vascular development" and "contractile fiber/actin cytoskeleton in myocytes".

## 3.4 GSEA

GSEA was performed to uncover the coordinated direction (activation or inhibition) of predefined gene sets at the whole-transcriptome level. In the CET group (Figs 4A-B), GO analysis revealed significant activation of pathways related to extracellular matrix remodeling and collagen fibril organization, including "extracellular matrix structural constituent," "collagen trimer," "collagen-containing extracellular matrix," "external encapsulating structure," and "chromatin architectural organization." Conversely, inhibited terms encompassed "spliceosomal complex," "positive regulation of DNA-templated transcription initiation," "skeletal muscle adaptation," "cytosolic large ribosomal subunit," and "cytosolic ribosome." KEGG pathway analysis further indicated the downregulation of processes such as "ubiquitin proteasome-mediated protein degradation," "abnormal protein degradation in VCP-associated disease," "protein clearance via α-synuclein/SOD1," and "mRNA translation initiation."

For the RES group (Figs 4C-D), GO analysis showed significant activation of "collagen-containing extracellular matrix," "extracellular matrix structural constituent," "external encapsulating structure," "collagen fibril organization," and "collagen trimer." Inhibited terms were associated with "mRNA processing," "mRNA splice site recognition," "nuclear speck," "DNA-binding transcription activator activity," and "regulation of mRNA metabolic process." KEGG analysis identified activation of pathways related to axonal transport, including "aberrant huntingtin protein binding to retrograde axonal transport machinery," "reference retrograde axonal transport," "RAB7-regulated microtubule minus-end directed transport," and "ARL8-regulated microtubule plus-end directed transport," alongside activation of "mitochondrial uncoupling protein 1-mediated thermogenesis." The "translation initiation" pathway was downregulated.

In the END group (Figs 4E-F), GO analysis demonstrated significant activation of "extracellular matrix structural constituent," "collagen trimer," "collagen fibril organization," "external encapsulating structure organization," and "collagen-containing extracellular matrix." Inhibited terms included "maturation of three-prer rRNA transcript to constituent subunits," "maturation of small subunit rRNA," "cytosolic ribosome," "skeletal muscle adaptation," and "ribosomal structural constituent." KEGG analysis highlighted the activation of several signaling pathways: "Integrin FAK RAC," "GPCR PLCB ITPR," "CXCR4 GNB PLCB PKC," "Integrin Talin Vinculin," and "Integrin FAK CDC42" signaling. The "mRNA translation initiation" process was inhibited.

## 3.5 WGCNA

The WGCNA algorithm was employed to identify key gene modules exhibiting significant differences before and after exercise in each group. A scale-free network was constructed with a soft thresholding power β = 9 (scale-free topology fit index $R^2 = 0.9$; Figs 5A–B). Dynamic tree cutting was applied to the clustering dendrogram (Fig 5C), ultimately identifying 20 distinct gene modules. The module eigengene values and their statistical significance pre- and post-intervention for each group are shown in Fig 5D. In the CET group, the MEbrown module was significantly upregulated after exercise (eigengene value = 0.26, P = 0.002), whereas the MElightgreen module was significantly downregulated (eigengene value = −0.20, P = 0.011). In the END group, only the MEbrown module showed significant upregulation (eigengene

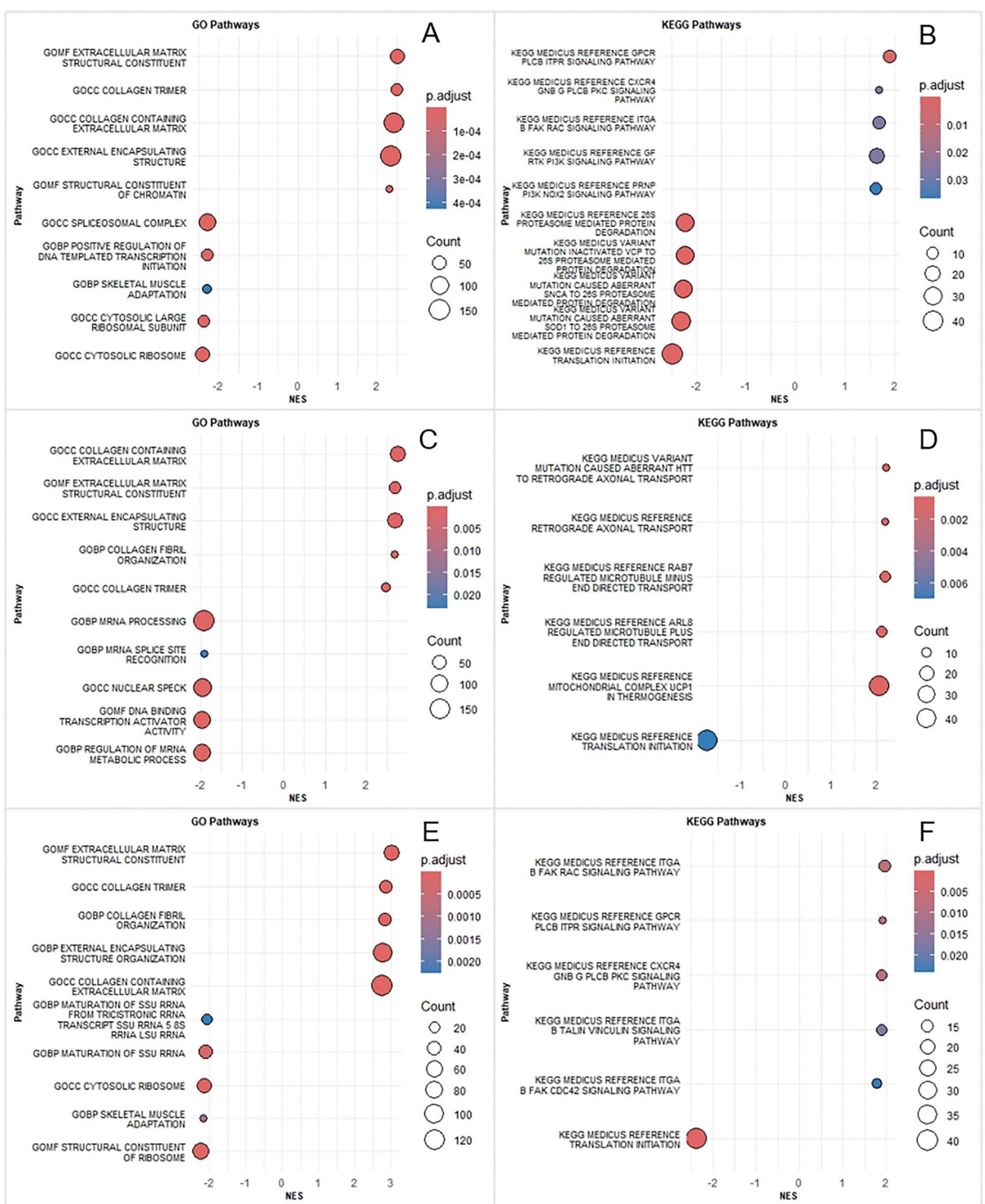

**Fig 4. GSEA bubble plots. (A-B)** CET group, **(C-D)** RES group, **(E-F)** END group. Each group displays up to five significantly activated and five inhibited pathways (fewer are shown if the threshold is not met). Bubble size corresponds to the number of enriched genes (Count) in the gene set. The color gradient represents the adjusted p-value (p.adjust), with red indicating higher significance and blue indicating lower significance. The horizontal axis shows the Normalized Enrichment Score (NES), where positive values denote pathway activation and negative values denote inhibition.

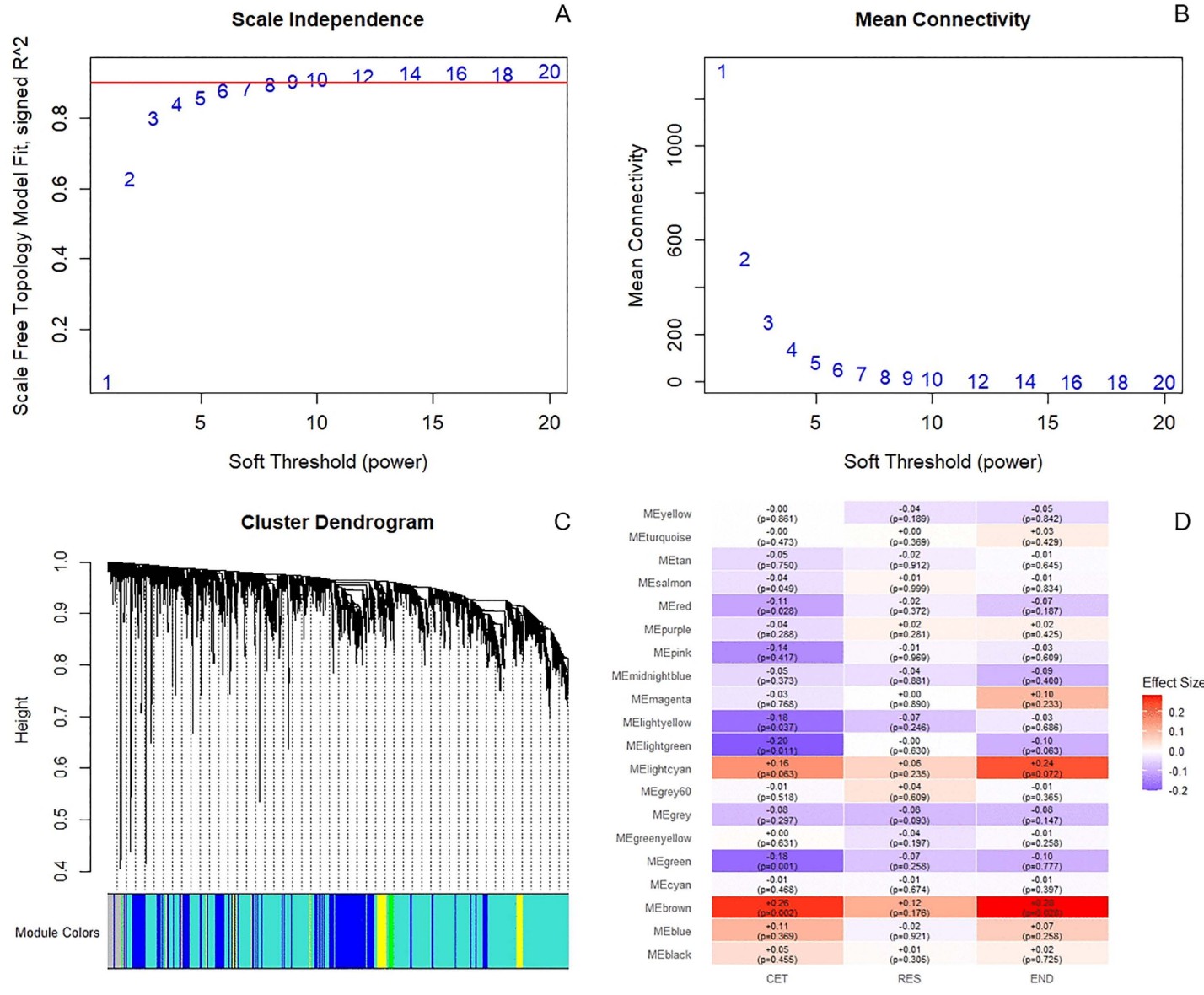

**Fig 5. Identification of differential modules by WGCNA pre- and post-exercise across groups. (A–B)** Soft threshold screening results: Based on the scale-free topology fit index and mean connectivity, a soft threshold power β = 9 was selected as the optimal value, achieving an approximate scale-free network topology (R² ≈ 0.9). **(C)** Gene co-expression clustering dendrogram and module assignment: Different colored bars below the dendrogram represent the 20 identified gene modules. **(D)** Heatmap displaying the difference in module eigengene (ME) values (Post − Pre) and their significance for each module across the CET, RES, and END groups. Color intensity represents the effect size (eigengene value change), with corresponding p-values indicated in parentheses.

value = 0.28, P = 0.026). No significant modules were detected in the RES group. It is noteworthy that the MEbrown module contained 576 genes.

## 3.6 Core gene identification

Genes from the significant modules identified by WGCNA were intersected with the differentially expressed genes (DEGs) from the CET and END groups, respectively, to obtain candidate key genes (Fig 6A). These candidate genes were then imported to construct and visualize protein-protein interaction (PPI) networks (Figs 6B–D). Within Cytoscape,

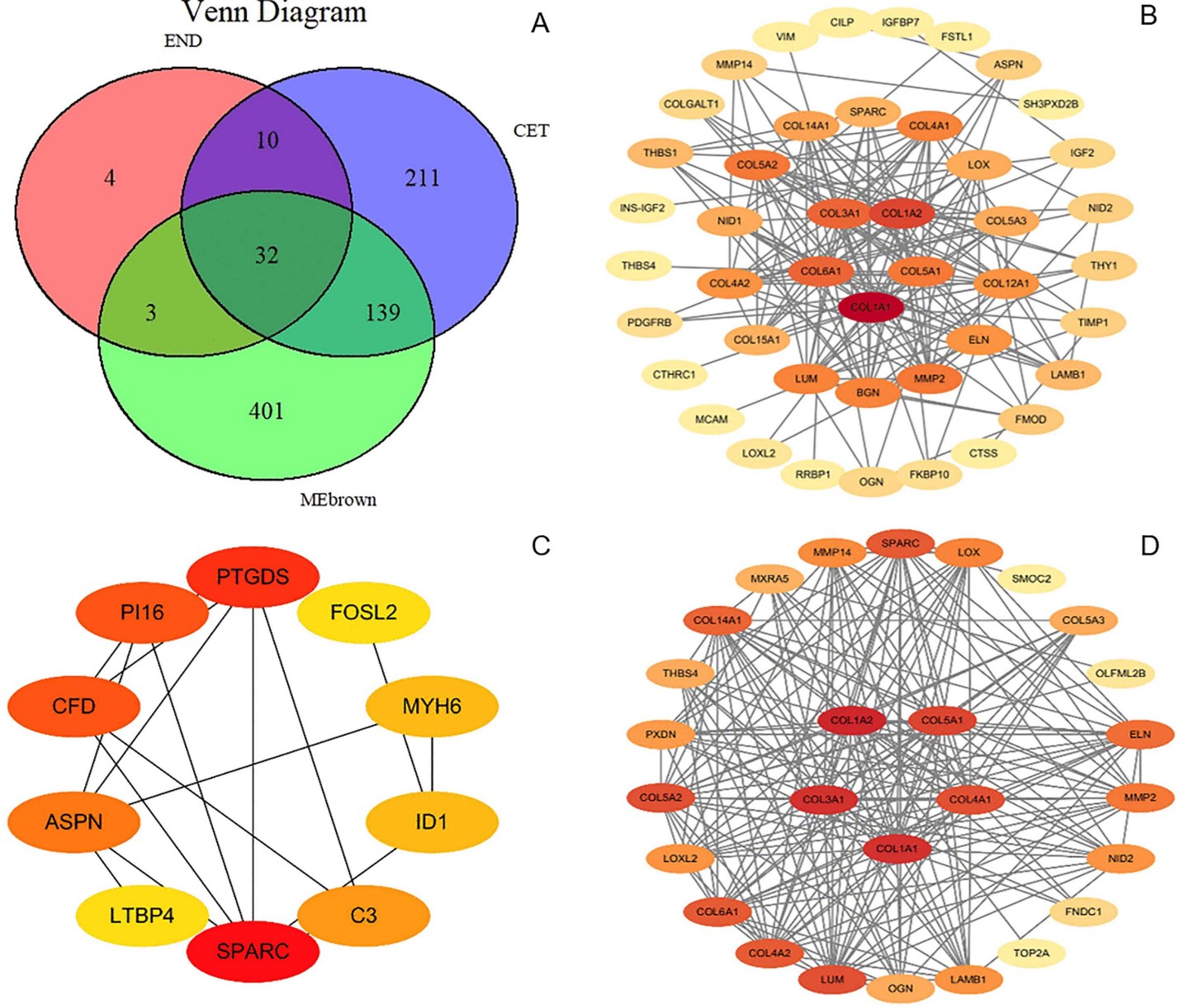

**Fig 6. Screening of candidate key genes and PPI network construction. (A)** Venn diagram illustrating the overlap between genes from the MEbrown module and the DEGs from the CET and END groups. **(B–D)** Topological layout of the PPI networks for candidate key genes in the **(B)** CET, **(C)** RES, and **(D)** END groups, respectively. Node color intensity corresponds to the Degree centrality, with red indicating higher degree and yellow indicating lower degree.

the cytoHubba plugin was used to score network nodes using five topological algorithms: MCC, Degree, MNC, EPC, and Closeness. For each algorithm, the top 10 ranked genes were extracted. Genes appearing in the top 10 lists of at least three algorithms were considered preliminary key candidates. The rankings from each algorithm were converted into scores (10 points for 1st, 9 for 2nd, …, 1 for 10th), and a composite score was calculated for each gene. The five genes with the highest composite scores were selected as the final key genes. The results showed that the core genes for the

CET group were COL1A1, COL1A2, COL3A1, COL6A1, and COL5A1, all of which were significantly upregulated post-exercise (Fig 7A). For the RES group, the core genes SPARC, PTGDS, and ASPN were significantly upregulated, while PI16 and CFD were downregulated (Fig 7B). In the END group, the core genes COL1A2, COL1A1, COL3A1, COL5A1, and COL4A1 were also significantly upregulated post-exercise (Fig 7C). Finally, the expression correlations of these key genes within their respective groups were analyzed and visualized to validate their co-expression patterns within the modules (Figs 7D–F).

### 3.7 TF-mRNA-miRNA regulatory network analysis

Regulatory networks for the key genes in each group were constructed using the NetworkAnalyst platform. As illustrated in Fig 8, the resulting networks reveal distinct regulatory patterns across different exercise modalities. In the CET group (Fig 8A), members of the miR-29 family (hsa-miR-29a, hsa-miR-29b, and hsa-miR-29c) were predicted to collectively

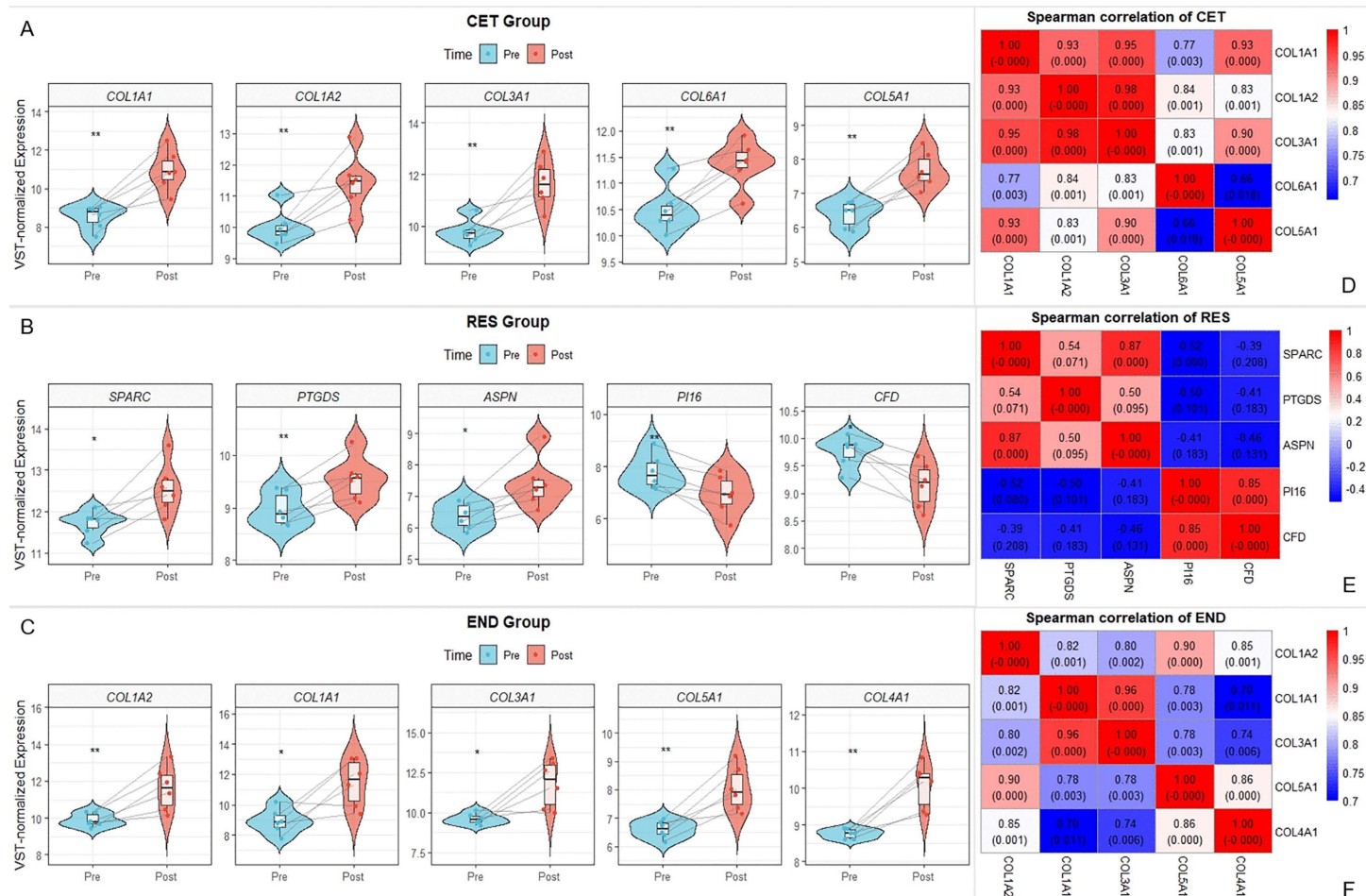

**Fig 7. Expression levels and correlations of core genes across groups. (A–C)** Comparison of VST-normalized relative expression levels for the key genes before (Pre) and after (Post) exercise in the **(A)** CET, **(B)** RES, and **(C)** END groups, respectively. Each subpanel combines a violin plot with an overlaid boxplot, and individual lines connect paired measurements from the same subject. Statistical significance was assessed using the paired Wilcoxon test. **(D–F)** Spearman correlation heatmaps illustrating pairwise correlations among these core genes within the **(D)** CET, **(E)** RES, and **(F)** END groups. The correlation coefficient (r) is displayed above each cell, with the corresponding p-value indicated in parentheses below. The background color intensity reflects the strength of the correlation. *$P < 0.05$, **$P < 0.01$.

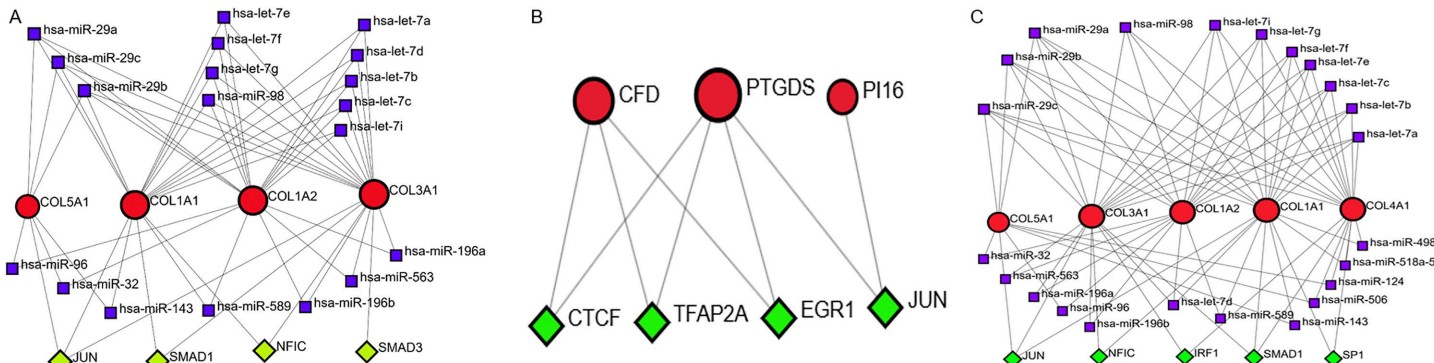

**Fig 8. TF–mRNA–miRNA regulatory networks. (A-C)** Networks for the CET, RES, and END groups, respectively. Red nodes represent mRNAs, green nodes represent predicted transcription factors (TFs), and purple nodes represent regulatory miRNAs. Edges display predicted regulatory relationships between TFs and mRNAs, as well as between miRNAs and mRNAs.

target four key mRNAs: COL1A1, COL1A2, COL3A1, and COL5A1. Additionally, the transcription factor JUN was found to simultaneously regulate COL1A1, COL3A1, and COL5A1. For the RES group (Fig 8B), PTGDS appeared to be regulated by multiple transcription factors, including JUN and TFAP2A. The regulatory network of the END group (Fig 8C) demonstrated that all five core mRNAs were co-regulated by the miR-29 family (hsa-miR-29a, hsa-miR-29b, and hsa-miR-29c), while the transcription factor JUN again targeted COL1A1, COL3A1, and COL5A1.

### 3.8 Muscle fiber proportion prediction

The proportions of slow-twitch and fast-twitch fibers in each sample were estimated by deconvoluting the sample expression matrix using a skeletal muscle signature matrix via the DeconRNASeq package (Fig 9A–C). The results showed that in the CET group, the pre-training slow-twitch fiber proportion was $0.455 \pm 0.032$ and the fast-twitch fiber proportion was $0.545 \pm 0.032$; post-training, these values were $0.435 \pm 0.070$ and $0.565 \pm 0.070$, respectively. In the RES group, the pre-training slow-twitch and fast-twitch proportions were $0.440 \pm 0.047$ and $0.560 \pm 0.047$, respectively, changing to $0.449 \pm 0.040$ and $0.551 \pm 0.040$ after training. For the END group, the pre-training proportions were $0.445 \pm 0.048$ for slow-twitch and $0.555 \pm 0.048$ for fast-twitch fibers, which became $0.427 \pm 0.063$ and $0.573 \pm 0.063$ post-training. No statistically significant differences were observed in the changes of slow-twitch or fast-twitch fiber proportions before and after training in any of the three groups ($p > 0.05$).

## 4 Discussion

This study employed a multi-faceted bioinformatics approach to systematically compare the effects of concurrent, resistance, and endurance training modalities on the skeletal muscle transcriptome in healthy young men. Our analyses reveal that despite their distinct physiological emphases, these training modes elicit adaptive responses at the transcriptomic level that share remarkable commonalities while also exhibiting distinct specificities.

Physiological data (Table 2) confirm that the 12-week intervention successfully induced mode-specific adaptations. First, all three groups exhibited comparable increases in lean body mass, suggesting that while muscle hypertrophy was universal, the functional quality of the muscle diverged according to training mode: the RES group demonstrated superior improvements in maximal strength, whereas the END group excelled in aerobic capacity. Notably, maximal muscle strength (1RM Leg Press) in the CET group was not compromised compared to the RES group. This preservation of strength likely results from the 24-hour recovery interval between sessions, which mitigates residual fatigue [26–28]. While the literature reports conflicting findings regarding the impact of concurrent training on strength development, these

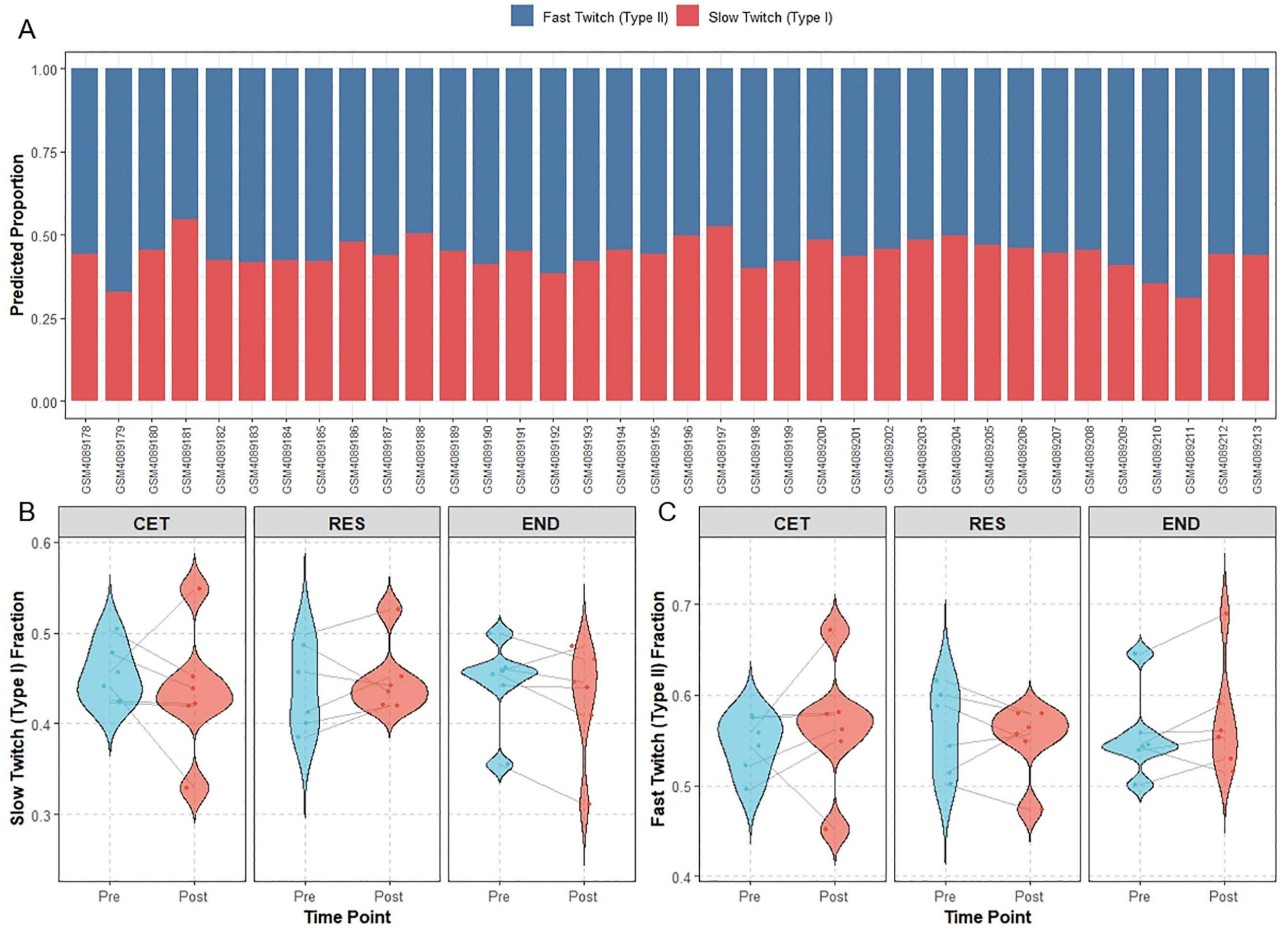

**Fig 9. Muscle fiber composition. (A)** Stacked bar plot showing the relative proportions of muscle fiber types (slow-twitch and fast-twitch fibers) for all samples. Each bar represents one sample, with the total height normalized to 100%. The colored segments within each bar indicate the percentage of slow-twitch (blue) and fast-twitch (red) fibers. **(B–C)** Violin plots displaying the paired changes in (B) slow-twitch and (C) fast-twitch fiber proportions before and after intervention in the three groups (CET, RES, END). The outlines show the distribution of fiber percentage values within each group, while lines connect paired measurements (pre- vs. post-intervention) from the same subject. Statistical significance was assessed using the paired Wilcoxon test.

discrepancies are generally attributable to differences in training volume, intensity, and frequency [29–34]. However, a distinct "interference effect" emerged in anaerobic capacity, specifically regarding explosive power development; the gain in Wingate peak power was significantly attenuated in the CET group compared to RES. This dissociation—where maximal strength is preserved but explosive power is blunted—suggests that concurrent training may specifically compromise the rate of force development (RFD) or velocity-specific adaptations [35–37], even in the presence of muscle growth.

In contrast to the divergent physiological outcomes, the most prominent transcriptomic finding was the remarkable convergence of adaptive signatures across all three groups. Differential expression analysis and GSEA consistently demonstrated robust activation of extracellular matrix (ECM) remodeling pathways, including "collagen trimer," "basement membrane," and "ECM-receptor interaction." This universal molecular signature parallels the physiological observation of increased Lean Body Mass across all groups. This aligns with the evolving understanding that the ECM is not merely a passive structural scaffold but a dynamic, mechanosensitive tissue crucial for transmitting mechanical signals [38,39]. Exercise-induced ECM remodeling is considered vital for optimizing myotendinous junctions, strengthening connective

tissue, and ultimately enhancing the mechanical properties of muscle [40,41]. This process holds particular significance for musculoskeletal health, especially considering that musculoskeletal injuries account for approximately 25%−30% of all years lived with disability in the elderly population and are often associated with poor healing outcomes [42,43]. In summary, our data strongly position ECM remodeling as a core adaptive mechanism in skeletal muscle's response to exercise [44,45].

Beyond the shared ECM response, specific pathways provided potential insights into the distinct functional adaptations, although the link between transcriptomics and the observed "interference effect" appears complex. Contrary to the hypothesis that concurrent training induces excessive catabolism, the CET group exhibited downregulation of the "ubiquitin-proteasome degradation" pathway and significant suppression of MSTN (Myostatin), a negative regulator of muscle mass. Concurrently, pathways related to "translation initiation" and "ribosome" were downregulated. It is crucial to note that the downregulation of translation-related pathways was not unique to CET but was also observed in RES and END groups (GSEA results). This likely reflects a specific temporal snapshot of the transcriptome (e.g., a post-exercise refractory period) rather than a chronic suppression of synthesis. The absence of a unique "catabolic" transcriptomic profile in the CET group suggests that the physiological blunting of explosive power (Wingate) may not be driven by gross muscle wasting or transcriptional inhibition of hypertrophy. Instead, the interference may stem from other mechanisms not captured by bulk transcriptomics, such as neuromuscular fatigue, alterations in motor unit discharge rates, or differential protein isoform expression [37,46–51].

The RES group specifically activated pathways associated with "retrograde axonal transport." Since resistance training adaptations rely heavily on neural factors, including motor unit recruitment and firing rates, the activation of axonal transport machinery provides a plausible molecular correlate to the superior strength improvements observed in this group [52,53]. The END group specifically activated "integrin signaling pathways" (e.g., Integrin-FAK-RAC). Integrins serve as mechanotransducers that link the ECM to intracellular signaling, often mediating capillary formation and metabolic adaptation [54–58]. This aligns with the END group's superior improvement in aerobic capacity.

Through WGCNA and subsequent PPI network analysis, we further identified core genes central to each training response. Notably, collagen-encoding genes such as COL1A1, COL1A2, and COL3A1 emerged as central nodes in both the CET and END groups, reinforcing the pivotal role of ECM constituents in exercise adaptation [40]. The upregulated expression of non-collagen ECM regulators (e.g., LUM, OGN, LOX, ASPN, SPARC) is critical for imparting flexibility and elasticity to the skeletal muscle collagen network [59,60]. The concomitant upregulation of the matrix metalloproteinase MMP14 suggests that this process represents an active "synthesis-degradation" remodeling rather than mere matrix accumulation [61,62]. Furthermore, the decreased expression of MSTN, a negative regulator of muscle growth, may potentially alleviate inhibition on satellite cell activation and muscle fiber growth. Within the RES group's core genes, ASPN—also upregulated in CET—functions as a regulator of TGF-β signaling involved in collagen deposition [63,64]. Similarly, SPARC influences the mechanical properties of the ECM by mediating cell-matrix interactions [65,66]. The upregulation of PTGDS hints at a role for prostaglandin metabolism in modulating the tissue microenvironment post-resistance training, potentially influencing tissue repair processes by fine-tuning local inflammatory balance [67]. The coordinated downregulation of PI16 and CFD may reflect a precise modulation of the repair microenvironment in skeletal muscle following resistance training [68,69].

Regulatory network analysis suggested that the miR-29 family and the transcription factor JUN might co-regulate key ECM genes. Existing literature indicates that miR-29 acts as a post-transcriptional regulator of collagen synthesis and is involved in maintaining ECM homeostasis [70,71]. JUN, a component of the AP-1 transcription factor complex, can be activated by mechanical stress and regulates processes like collagen synthesis by binding to AP-1 response elements in target gene promoters [72–74]. Our prediction that miR-29 and JUN may target overlapping sets of collagen genes suggests a potential balancing mechanism in exercise-induced ECM remodeling: mechanical stress might promote collagen gene transcription via JUN, while miR-29 could prevent excessive ECM accumulation by degrading collagen mRNAs. These two mechanisms might work in concert to fine-tune the dynamic balance between ECM synthesis and degradation.

Finally, the absence of significant changes in muscle fiber type composition after the 12-week intervention aligns with findings from some human training studies [75]. This indicates that fiber type transformation might be a more protracted process, or that early adaptations primarily involve functional optimization within existing fibers rather than a wholesale switch in myosin heavy chain isoforms, even while widespread transcriptomic adaptations are already underway.

We acknowledge several limitations in this study. As a bioinformatics investigation, our findings are primarily derived from computational analyses of a public dataset and require subsequent experimental validation. The relatively limited sample size might have constrained the statistical power to detect more subtle changes. Furthermore, the estimation of muscle fiber proportions via deconvolution algorithms relies on a predefined gene signature and is inherently less precise than biochemical analyses of myosin isoform expression. These limitations necessitate a cautious interpretation of our conclusions and highlight directions for more comprehensive future research.

## 5 Conclusion

In summary, this integrated multi-level transcriptomic analysis, substantiated by physiological outcomes, systematically reveals that concurrent, resistance, and endurance training share a common adaptive signature—the induction of skeletal muscle ECM remodeling—which parallels the universal increase in lean body mass. While exhibiting distinct regulatory profiles through the proteasome degradation, retrograde axonal transport, and integrin signaling pathways, respectively, physiological assessments confirmed mode-specific trade-offs: the END group showed the poorest strength gains, the RES group lacked aerobic improvements, and the CET group, despite preserving maximal strength, exhibited a blunted capacity for explosive power development. Our findings not only identify core ECM genes, including COL1A1 and SPARC, and propose a potential miR-29/JUN regulatory axis, but also demonstrate that short-term training induces profound molecular adaptations prior to observable changes in muscle fiber type composition. Collectively, these insights provide an important molecular theoretical foundation for understanding the regulatory mechanisms of exercise adaptation and for informing the development of precise exercise prescriptions.

## Supporting information

**S1 Table. Training program.**
(XLSX)

**S2 Table. Physiological data.**
(XLSX)

## Author contributions

**Conceptualization:** Longfei Zhao, Huangyan Li, Li luo, Shiliang Hu.

**Data curation:** Longfei Zhao, Huangyan Li.

**Formal analysis:** Longfei Zhao, Huangyan Li, Dongli Li.

**Funding acquisition:** Li luo, Shiliang Hu.

**Investigation:** Longfei Zhao, Huangyan Li, Dongli Li.

**Methodology:** Longfei Zhao, Huangyan Li.

**Project administration:** Li luo, Shiliang Hu.

**Software:** Longfei Zhao, Huangyan Li.

**Supervision:** Li luo, Shiliang Hu.

**Validation:** Longfei Zhao, Huangyan Li, Dongli Li, Li luo, Shiliang Hu.

**Visualization:** Longfei Zhao, Huangyan Li, Dongli Li.

**Writing – original draft:** Longfei Zhao, Huangyan Li, Shiliang Hu.

**Writing – review & editing:** Longfei Zhao, Huangyan Li, Dongli Li, Li luo, Shiliang Hu.

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
