## [Decision Letter · Decision Letter 0]

25 Nov 2025

Dear Dr. Hu,

Thank you for submitting your manuscript to PLOS ONE. After careful consideration, we feel that it has merit but does not fully meet PLOS ONE’s publication criteria as it currently stands. Therefore, we invite you to submit a revised version of the manuscript that addresses the points raised during the review process.

We look forward to receiving your revised manuscript.

Kind regards,

Donald Gullberg, PhD

Academic Editor

PLOS ONE

Journal Requirements:

“This work was supported by University Humanities and Social Sciences Research Project of Department of education of Guizhou Province (24RWZX026); High-Level Talent Research Start-Up Fund Project of Guizhou Medical University (2024008).”

“The author(s) received no specific funding for this work”

“The author(s) received no specific funding for this work”

5. We note that your Data Availability Statement is currently as follows: “All relevant data are within the manuscript and its Supporting Information files”

6. PLOS requires an ORCID iD for the corresponding author in Editorial Manager on papers submitted after December 6th, 2016. Please ensure that you have an ORCID iD and that it is validated in Editorial Manager. To do this, go to ‘Update my Information’ (in the upper left-hand corner of the main menu), and click on the Fetch/Validate link next to the ORCID field. This will take you to the ORCID site and allow you to create a new iD or authenticate a pre-existing iD in Editorial Manager.

Reviewers' comments:

Reviewer's Responses to Questions

**Comments to the Author**

1. Is the manuscript technically sound, and do the data support the conclusions?

Reviewer #1: Yes

2. Has the statistical analysis been performed appropriately and rigorously?

Reviewer #1: Yes

3. Have the authors made all data underlying the findings in their manuscript fully available?

Reviewer #1: Yes

4. Is the manuscript presented in an intelligible fashion and written in standard English?

Reviewer #1: Yes

Reviewer #1: In the ms. entitled ”Transcriptomic analysis reveals the impact of concurrent, resistance, and endurance training on skeletal muscle”, Zhao and co-workers have taken a transcriptomic approach to unravel common and separate transcriptomic mechanisms underlying different modes of physical training. It is shown that extracellular matrix remodeling is common to concurrent, resistance, and endurance training while distinct training-specific differences were also observed in gene expression between the three modes of training.

General comments: This is an interesting and ambitious approach to unravel differences in the gene expression response to three different modes of physical exercise regimes. In study limitations it is advised that changes in gene expression give important information on changes at the mRNA level which need to be translated to the protein level. In addition, there are other concerns which need to be addressed (see specific comments below).

Specific comments: The subjects participating in the study need to be described in more detail (age, body weight, height, baseline training status etc).

Concurrent, resistance and endurance training mean different for different people and these types of trainings need to be presented in detail.

The response to the different training regimes should be presented such as changes in body weight, strength and maximal oxygen uptake.

Minor comment: As mentioned at the end of the Discussion in study limitations, transcriptomics are associated with numerous limitations in the analyses of muscle fiber type transitions in response to e.g. physical exercise. However, enzyme-histochemical stainings are also associated with significant limitations. It is suggested that histochemistry is replaced by biochemical analyses of myosin isoform expression.

**Do you want your identity to be public for this peer review?** For information about this choice, including consent withdrawal, please see our Privacy Policy

Reviewer #1: No

---

## [Author Response · Author response to Decision Letter 1]

3 Dec 2025

Journal Requirements:

1. Style Requirements: We have reformatted the manuscript using the provided PLOS ONE style templates.

2. & 3. & 4. Funding Statement: We have removed all funding-related text from the manuscript. Our updated funding declaration is as follows, and we kindly request you update the online submission form accordingly:

This study was supported by the University Humanities and Social Sciences Research Project of the Department of Education of Guizhou Province (Grant No. 24RWZX026) and the High-Level Talent Research Start-Up Fund Project of Guizhou Medical University (Grant No. 2024008). Author Longfei Zhao is the recipient of both grants. The funders had no role in study design, data collection and analysis, decision to publish, or preparation of the manuscript.

3. Data Availability Statement: We confirm our submission contains the minimal data set to replicate all findings. The primary gene expression dataset analyzed is the publicly available GEO dataset GSE137832. All other data are provided within the manuscript and its Supporting Information files.

4. ORCID iD: We confirm that the corresponding author’s ORCID iD has been validated in Editorial Manager.

Response to Reviewer #1:

We thank the reviewer for their positive and constructive feedback.

• Subject Details, Training Protocols, and Physiological Outcomes: As suggested, we have now included a detailed description of the participants' baseline characteristics (age, height, body mass, BMI), a concise specification of each supervised 12-week training protocol, and the key physiological outcomes (changes in lean body mass, strength, aerobic capacity, anaerobic power) in the revised manuscript (Methods and Results sections).

• Limitations - Biochemical Analysis: We appreciate this insightful comment. The relevant sentence in the Limitations section has been amended from "direct histochemical staining" to "biochemical analysis of myosin isoform expression."

All changes have been made in the revised manuscript file. We believe these revisions have strengthened the manuscript and thank the editor and reviewer for their time and valuable input.

---

## [Editor Report · Decision Letter 1]

18 Dec 2025

Transcriptomic Analysis Reveals the Impact of Concurrent, Resistance, and Endurance Training on Skeletal Muscle

PONE-D-25-54953R1

Dear Dr. Hu,

We’re pleased to inform you that your manuscript has been judged scientifically suitable for publication and will be formally accepted for publication once it meets all outstanding technical requirements.

Kind regards,

Donald Gullberg, PhD

Academic Editor

PLOS One

Additional Editor Comments (optional):

An editor has reviewed the revised version and finds that the MS is a well written and interesting study of the effects of training on muscle gene transcription. The study opens up for more empirically based studies where gene expression in different cell types in different microenvironmental niches are studied in more detail.
---

## [Editor Report · Acceptance letter]

PONE-D-25-54953R1

PLOS One

Dear Dr. Hu,

I'm pleased to inform you that your manuscript has been deemed suitable for publication in PLOS One. Congratulations! Your manuscript is now being handed over to our production team.

Kind regards,

on behalf of

Professor Donald Gullberg

Academic Editor

PLOS One